# BENCHMARKS FOR REINFORCEMENT LEARNING WITH BIASED OFFLINE DATA AND IMPERFECT SIMULATORS

## ABSTRACT

In many reinforcement learning (RL) applications one cannot easily let the agent act in the world; this is true for autonomous vehicles, healthcare applications, and even some recommender systems, to name a few examples. Offline RL provides a way to train agents without exploration, but is often faced with biases due to data distribution shifts, limited exploration, and incomplete representation of the environment. To address these issues, practical applications have tried to combine simulators with grounded offline data, using so-called *hybrid methods*. However, constructing a reliable simulator is in itself often challenging due to intricate system complexities as well as missing or incomplete information. In this work, we outline four principal challenges for combining offline data with imperfect simulators in RL: simulator modeling error, partial observability, state and action discrepancies, and hidden confounding. To help drive the RL community to pursue these problems, we construct "Benchmarks for Mechanistic Offline Reinforcement Learning" (B4MRL), which provide dataset-simulator benchmarks for the aforementioned challenges. Finally, we propose a new approach to combine an imperfect simulator with biased data and demonstrate its efficiency. Our results suggest the key necessity of such benchmarks for future research.

## 1 INTRODUCTION

Reinforcement learning (RL) is a learning paradigm in which an agent explores an environment in order to maximize a reward Sutton and Barto (2018). However, in many applications exploration can be costly, risky, slow, or impossible due to legal or ethical constraints. These challenges are evident in fields such as healthcare, autonomous driving, and recommender systems.

To overcome these obstacles, two principal methodologies have emerged: using offline data, and incorporating simulators of real-world dynamics. Both approaches have distinct advantages and drawbacks. While offline data is sampled from real-world dynamics and often represents expert policies and preferences, it is limited by exploration and finite data Levine et al. (2020); Fu et al. (2020); Jin et al. (2021). Furthermore, offline data often suffers from confounding bias, which occurs when the agent whose actions are reflected in the offline dataset acted based on information not fully present in the available data: For example, a human driver acting based on eye-contact with another driver, or a clinician acting based on an unrecorded visual inspection of the patient. Confounding can severely mislead the learning agent Zhang and Bareinboim (2016); Gottesman et al. (2019); De Haan et al. (2019); Wang et al. (2021), as we demonstrate in our paper. We refer to these sources of error as *offline2real*.

In contrast to learning from offline data, simulators allow nearly unlimited exploration, and have been the bedrock of several recent triumphs of RL (Mnih et al., 2013; Vinyals et al., 2019; Wang et al., 2023). However, utilizing simulators brings its own set of challenges, most notably – modeling error. This error often arises due to the complexity of real-world dynamics and the inevitability of missing or incomplete information. Although simplified simulators are widely used, any discrepancies between their dynamics and real-world dynamics can lead to unreliable predictions. These so-called *sim2real* gaps may range from misspecifications in the transition and action models to biases in the observation functions (Abbeel et al., 2006; Serban et al., 2020; Kaspar et al., 2020; Ramakrishnan et al., 2020; Arndt et al., 2020).

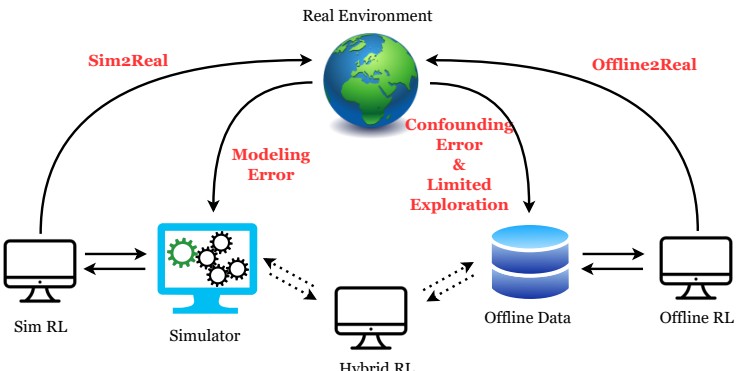

Figure 1: An illustration of the discrepancies and biases arising when training RL agents. *Modeling error* refers to the discrepancy between the real world dynamics and the simulator, e.g. transition error. *Confounding error* refers to bias due to the dataset not including factors affecting the behavioral policy. Other challenges include limited exploration, partial observability and state and action discrepancies, as detailed in Section 2.

In recent years, there has been a growing recognition of the complementary strengths and limitations of offline data and simulation-based approaches in RL (Nair et al., 2020; Song et al., 2022; Niu et al., 2022; 2023). Recent work has merged these two approaches to leverage their respective advantages and mitigate their drawbacks; namely, offline data, which provides real-world expertise and preferences, with simulators which offer extensive exploration capabilities. These hybrid methods hold promise for addressing the challenges posed by costly or limited exploration in various domains.

In this work, we present and study four key challenges for merging simulation and offline data in RL: modeling error, partial observability, state and action discrepancies, and confounding error. We propose the first set of benchmarks to systematically explore hybrid RL approaches, which we term "Benchmarks for Mechanistic Offline Reinforcement Learning" (B4MRL). Each benchmark is driven by differences in the properties of simulation and offline data. We demonstrate how contemporary offline RL approaches can fail on these benchmarks, as well as failure modes for hybrid approaches, suggesting the key necessity of such benchmarks for future research. We further propose a baseline of our own, which learns and corrects for discrepancies between offline data and simulation, showing significant performance increase on some of our benchmarks. We hope the benchmarks will drive the RL community towards a better understanding of the challenges of using both offline data and imperfect simulators, helping the community build better, safer and more reliable models.

## 2 CHALLENGES OF COMBINING OFFLINE DATA WITH SIMULATORS

In this section, we outline some key challenges in merging simulation and offline data, which we partition into four somewhat overlapping categories: modeling error, partial observability and state discrepancies, action discrepancies, and confounding bias. Each of these challenges stems from differences and gaps between the true dynamics and either the simulator or the available offline data. Modeling error is due to the impossibility of exactly modeling real-world dynamics. Partial observability and state discrepancy deal with both the limitation of the simulator in encapsulating the entire observation space, and the limitations of recording information in real-world systems. Action discrepancy results from different levels of abstracting actions in simulators versus real-world data. Lastly, confounding bias, which is a special and important case of partial observability in offline data, addresses the specific problem of hidden factors influencing both the observed decisions and outcomes in the process that generated the offline data. Understanding and addressing these challenges is crucial for designing robust RL agents capable of transferring their learning from simulations to the real world (Sim2Real) and from offline data to the real world (Offline2Real). Finally, we note that offline data is inherently prone to errors due to limited exploration; as this has been shown by previous work (Levine et al., 2020; Fu et al., 2020; Jin et al., 2021), we do not focus on it in our analysis of benchmark results below. A general illustration of these challenges is depicted in Figure 1.

## 2.1 MODELING ERROR (SIM2REAL)

Simulators, as computational representations of real-world systems, inherently contain modeling errors. These errors arise from simplifications and assumptions made during the simulator's design and construction to render the simulation manageable and computationally tractable, a process which often introduces systemic differences or biases between the simulator's dynamics and the real-world system. For example, a weather simulation may be biased due to an imperfect understanding of atmospheric dynamics, and a diabetes simulation might not accurately simulate the complexities of the body's reaction to exercise. Consequently, these biases can influence the decisions and actions taken by a reinforcement learning agent trained on such simulators, leading to suboptimal performance when transferred to the real world.

## 2.2 PARTIAL OBSERVABILITY AND STATE DISCREPANCY (SIM2REAL, OFFLINE2REAL)

Simulators are often designed to abstract and simplify real-world complexities, selectively modeling aspects of a problem that are most relevant to the intended application. This selective modeling inadvertently creates blind spots, as parts of the real-world observation space are omitted or over-simplified. For example, consider an autonomous driving simulator. It might accurately model the dynamics of vehicles and pedestrian movement, crucial aspects for safe navigation. However, to keep the simulator manageable and tractable, it may exclude details such as subtleties of human behavior, including facial expressions or gestures that could signal an intent to cross the road. Despite these omissions, the simulator remains a valuable tool for training autonomous driving systems. However, its partial state description can lead to biases in the learned policy, which might be suboptimal or even erroneous in the real world.

State discrepancy is not limited to partial observability, and could also take place when the states are represented in a different way between the simulator and the real world. For example, the autonomous vehicle simulator's state holds the angle and velocity of the vehicle, but the real world environment states are the output of the vehicle's sensors, like a video camera, a LiDAR sensor etc.

Similarly, in Offline2Real scenarios, the data collected from real-world environments might suffer from partial observability. This could be due to constraints in the data collection process or limitations in sensor technology. For instance, in healthcare settings, electronic health records might not capture all relevant information about a patient's lifestyle, mental state, or genetic factors, which can significantly influence health outcomes. Partial observability in offline data may or may not lead to confounding bias, as we discuss later in Section 2.4.

## 2.3 ACTION DISCREPANCY (SIM2REAL)

One of the substantial challenges in merging simulation and offline data lies in inconsistencies between action definitions in simulation environments and offline data. Every action taken by an agent in the real environment can be nuanced and multifaceted. Simulators, on the other hand, have to abstract these complexities into a more manageable and computationally feasible representation. As a result, there can be a disconnect in how actions are represented in these two different systems.

For example, in an autonomous driving system, the action might be discrete and only choose between moving a lane to the left, a lane to the right or stay at the current lane. However, in real-world data, the actions might also include more specific information like the exact amount of torque change, and the steering angle.

Furthermore, real-world time delays in action execution might not be mirrored in simulators, adding another layer of complexity. For instance, in a real-world driving scenario, there is a brief delay between when a driver decides to apply the brakes and when the car actually begins to slow down due to the physical process involved. However, a simulator might implement the braking action instantaneously, leading to discrepancies when combining simulation and offline data.

## 2.4 CONFOUNDING BIAS (OFFLINE2REAL)

The presence of unobserved (hidden) confounding variables poses a significant challenge when using observational data for decision-making. Hidden confounding occurs when in the process that

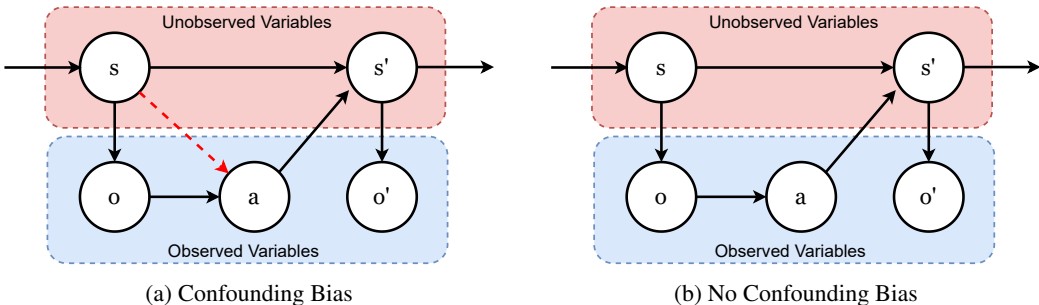

Figure 2: Both figures represent the causal graph of a POMDP. While in both cases the state $s$ is not observed, only in figure (a) $s$ acts as confounder, as actions in the data were taken w.r.t. the unobserved $s$.

generated the offline data, unobserved factors exerted influence on both the outcome and the decisions made by the agent. This can lead to unbounded bias, a result which is well known from the causal inference literature Pearl (2009); Zhang and Bareinboim (2016); Tennenholtz et al. (2020; 2022); Uehara et al. (2022); Hong et al. (2023). This issue becomes particularly pertinent in sequential decision-making scenarios and can substantially impact the performance of learned policies. Hidden confounding is prevalent in diverse real-world applications, including autonomous driving, where unobserved factors like road conditions affect the behavior of the human driver, and healthcare, where for example unrecorded patient information or patient preferences may influence the decisions made by physicians as well as patient outcome. A plot depicting hidden confounders in POMDPs is shown in Figure 2.

Effectively addressing hidden confounding in offline RL is paramount to ensure the reliability and effectiveness of learned policies. Research has attempted to develop methodologies to account for confounding bias, including: the identification of hidden confounders (Angrist et al., 1996; Jaber et al., 2018; Lee and Bareinboim, 2021; von Kügelgen et al., 2023), using interventions or additional data sources (Kallus et al., 2018; Zhang and Bareinboim, 2019; Tennenholtz et al., 2021; Lee et al., 2020), and the quantification and integration of uncertainty arising from confounding into the learning process (Pace et al., 2023). There is a crucial need for benchmarks and datasets specifically designed to address this issue, enabling researchers to compare and evaluate different methods for handling confounding bias in offline RL. We emphasize that hidden confounding and partial observability are distinct concepts. While they intersect in some cases, it is crucial to recognize their differences to effectively address their challenges, as we demonstrate in the following example.

To demonstrate the impact of hidden confounding bias in offline RL, consider the following single-state decision problem with two actions $\{a_0, a_1\}$. We let $z \in \{0, 1\}$ such that $P(z = 0) = \frac{1}{3}$, and $P(z = 1) = \frac{2}{3}$. Additionally, let the reward $r \in \{0, 1\}$, such that $P(r = 1 | z = 1, a = a_1) = \frac{1}{2}$, $P(r = 1 | z = 1, a = a_0) = \frac{1}{3}$, $P(r = 1 | z = 0, a = a_1) = \frac{1}{4}$ and $P(r = 1 | z = 1, a = a_0) = \frac{1}{6}$. Note that action $a_1$ dominates, and with or without access to $z$ at decision time the optimal action is given by $a^* = a_1 = \arg\max_a \mathbb{E}_{z \sim P(z)} P(r = 1 | z, a)$.

Next, let $\pi_b(a|z)$ be some behavioral policy (with access to $z$), which deterministically selects action $a_1$ when $z = 0$ or selects action $a_0$ when $z = 1$. We ask, can data generated by $\pi_b$ be used to learn a good policy if $z$ is not provided in the data? That is, can we learn a policy which maximizes $\mathbb{E}_{z \sim P(z)}[P(r = 1 | a, z)]$? Unfortunately, $z$ acts as a hidden confounder, which significantly biases our results, even in the limit of infinite data. Indeed, our data is sampled from $P^{\pi_b}(r, a) = \mathbb{E}_{z \sim P(z)}[P(r | a, z)\pi_b(a|z)]$, and thus $P^{\pi_b}(r = 1 | a = a_0) = \frac{\mathbb{E}_{z \sim P(z)}[P(r=1|a_0,z)\pi_b(a_0|z)]}{\mathbb{E}_{z \sim P(z)}[\pi_b(a_0|z)]} = P(r = 1 | a_0, z = 1) = \frac{1}{3}$. Similarly, $P^{\pi_b}(r = 1 | a = a_1) = P(r = 1 | a_1, z = 0) = \frac{1}{4}$. Therefore, even in the limit of infinite data, the standard empirical estimator $\hat{\pi} \in \arg\max_a P^{\pi_b}(r = 1 | a)$ would yield a suboptimal result of selecting action $a_0$. Notice that this error is due to the dependence of both $a$ and $r$ on the hidden confounder $z$, and not only by the fact that it is unobserved. Moreover, this bias cannot be mitigated with increasing the number of samples, unlike the statistical uncertainty induced by finite data.

In the next section, we shift our focus to developing benchmarks that serve as a rigorous testing ground for RL algorithms. These benchmarks were designed to illuminate the aforementioned challenges, helping researchers devise strategies to mitigate them, thereby promoting the advancement of robust, reliable, and high-performing RL systems that effectively utilize both offline data and simulations.

# 3 BENCHMARKS FOR MECHANISTIC OFFLINE REINFORCEMENT LEARNING (B4MRL)

In this section, we outline the "Benchmarks for Mechanistic Offline Reinforcement Learning" (B4MRL), designed for evaluation of RL methods using both offline data and simulators, which we refer to as hybrid algorithms. The proposed datasets and simulators encompass a range of discrepancies between the true dynamics, the simulator, and the observed data.

Given the four principal challenges delineated in Section 2 – namely, modeling error, partial observability, discrepancies in states and actions, and confounding bias – we created a benchmark rooted in the MuJoCo robotic environment (Todorov et al., 2012), and the Highway environment (Leurent, 2018). The MuJoCo tasks are popular benchmarks used to compare both offline RL and online RL algorithms, including multiple environments: HalfCheetah, Hopper, Humanoid, Walker2D. These environments provide the agent observations of various variables of the controlled robot, such as the angle and angular-velocity of the robot joints, and the position and velocity of the different robotic parts (e.g., an observation in HalfCheetah consists of 17 variables). The acting agent can perform actions at a given time by applying different torques to each joint (e.g. in HalfCheetah there are 6 joints, hence an action consists of 6 continuous variables). The reward function differs between the different tasks, and relies mainly on the speed and balance of the robot. The Highway environment simulates the behavior of a vehicle that aims to maintain high speed while avoiding collisions. This environment provides the agent observations that include the current position and velocity of the controlled vehicle and the other vehicles on the road, and lets the agent control the throttle and steering angle of the controlled vehicle.

In recent years several MuJoCO-based offline-RL benchmarks and datasets emerged, offering different characteristics and challenges. The most common one, and the one we build upon in this paper, is the Datasets for Deep Data-Driven Reinforcement Learning benchmark, or D4RL (Fu et al., 2020). These datasets are categorized by scores achieved by an underlying data-generating-agent, ranging from completely random agents, to "medium" level agents, through expert agents, and further provide datasets with heterogeneous policy mixtures (e.g., medium-expert). We note that by construction, these datasets do not suffer from hidden confounding. Our work builds upon and expands these datasets by implementing imperfect simulators and the other challenges outlined in Section 2. While the aim of this paper is to provide benchmarks for hybrid-RL algorithms, we stress that the benchmarks we provide in some of the challenges could also be used to test offline-RL and online-RL algorithms. We constructed these benchmarks such that researchers can easily create new benchmarks for evaluating the various challenges. We refer the reader to Appendix A for exhaustive details.

**Challenge 1: Modeling Error.** We induce modeling error by introducing changes in simulator dynamics which directly influence the transition function over time. Small errors in transition dynamics could aggregate to produce completely wrong state predictions over long horizons. Specifically, in this benchmark we propose changing one of the environment parameters that has an effect on the simulator's dynamics. For example, in the HalfCheetah environment, we propose two benchmarks: changing the gravitation parameter to $g_{\text{sim}} = 19.6$ instead of $9.81$, and changing the friction parameter by multiplying it by a factor of $0.3$.

**Challenge 2: Partial Observability and State Discrepancy.** We induce this challenge in two ways: (1) some of the variables in the full state are hidden from the online algorithm, and (2) the observations available to the online algorithm are a noisy version of the true environment states, being observed with added Gaussian noise. For (1) we provide two benchmarks for removing a variable from the observations, with two levels of effect on the simulator: low effect (named $h_{\text{low}}$), and high effect (named $h_{\text{high}}$). We chose two specific features after benchmarking the Soft Actor-Critic (SAC) (Haarnoja et al., 2018) algorithm against all features (i.e., removing each possible single feature). We demonstrate the different effect of removing variables from the observation of the simulator

on HalfCheetah in Figure 4. For (2), we used two noise levels: noise with low variance $\sigma_{\text{low}}$ and noise with high variance $\sigma_{\text{high}}$. Combining these options with available offline datasets yields 16 benchmarks: four simulator discrepancy times four D4RL datasets (random, medium, medium-replay, and medium-expert).

In addition to benchmarks with partial observability in the simulator, we add a complementary benchmark with partial observability in the dataset. This is achieved by creating a new dataset with a data generating agent that trains and collects data on partially observed states of the environment. To form this benchmark we created two new datasets, with a different missing variable in each. Specifically, we removed the same variables $h_{\text{low}}$ and $h_{\text{high}}$, as described above. For the hybrid-RL algorithms we combine the new datasets with a simulator that suffers from transition error, resulting in a total of two benchmarks. Importantly, while these datasets suffer from partial observability, they do not suffer from hidden confounding, as the data-generating agent decides on its next action based on the same observation that is registered in the data; see Figure 2b.

**Challenge 3: Action Discrepancy.**    The third challenge centers around the issue of discrepancies between actions. To allow evaluation of the impact of action errors we altered how actions taken by the agent in the simulator state dictate the transition to the next state. To that end, we suggest two different scenarios: noisy actions, and delayed actions.

For noisy actions, we integrate Gaussian noise into the action implemented by the agent to the simulator's present state, whereas the dataset's actions remain without noise, creating a discrepancy between the simulator and the data in the effect of actions on the state. We benchmark the models on two noise levels: noise with low variance $\sigma_{\text{low}}$, and noise with high variance $\sigma_{\text{high}}$. As before, the choice of values was done based on the results of the SAC algorithm on the noisy simulator. The benchmark includes the combination of the 4 D4RL datasets and a simulator with action discrepancy (low noise, high noise) resulting in 8 different datasets.

For delayed actions, we add a delay between the time an action occured, and when it affects the state of the environment. To model action delay, we add a queue to the environment, and at each step, a new action is inserted to the queue. Each action has a random timer of when it can be taken, and for each benchmark we set the random timer to have a different mean $\mu$. We propose three benchmarks with different levels of delay: $\mu = 1, 2$ and $3$. The variance of the action is set to $1$, and the sampled delay is rounded to the closest integer (we do not let the action's timer drop below zero). The simulator in this case has no knowledge of the inherent delay, and always simulates the action delay as zero. Note that unlike the other benchmarks, here the ground-truth environment that is used for evaluation is the environment with action delay.

**Challenge 4: Confounding Bias.**    For this challenge, we assume we do not have complete access to the state that the data generating agent utilized when determining its actions. This is a special and important case of partial observability which occurs in offline data and can induce bias due to the behavior policy's dependence on the unobserved factors, see Section 2.4.

For this benchmark we build on the D4RL datasets as follows: We either add Gaussian noise to the observations in the data, or we omit a dimension recorded in the dataset observations. This is similar to the partial observability challenge described above, but here information used by the agent during the data generation process was removed from the dataset, and not from the simulator. Thus, the data generating agent decided on action $a_i$ based on the full system state $s_i$, but we have access only to a noisy or projected observation $o_i$; see Figure 2a. This creates a dataset with hidden confounding, where we do not have full information on why a specific action was chosen. Recall that, as mentioned in Section 2.4, learning from data with such hidden confounders might in general incur arbitrary levels of bias (Pearl, 2009; Zhang and Bareinboim, 2016; Tennenholtz et al., 2020; 2022; Uehara et al., 2022; Hong et al., 2023). We used the same settings as the the observation-error benchmark: low and high Gaussian noise on the observations in the data ($\sigma_{\text{low}}$ and $\sigma_{\text{high}}$), and missing dimensions ($h_{\text{low}}$ and $h_{\text{high}}$) from the observations in the data, resulting in 16 benchmarks.

Finally, we provide an additional benchmark for confounded datasets by creating a new dataset where the data-generating-agent observes and selects actions based on a history of three observations, instead of the last one. This scenario is similar to the dataset proposed in the partial observability challenge, but here the data-generating agent has taken its decisions based on a history of observations, and not the last observation alone. Hiding the fact that the dataset actions were history-aware can

induce hidden confounding. For this benchmark we create the history-aware dataset with hidden variables ($h_{\text{low}}$ and $h_{\text{high}}$), and use a simulator with transition error.

As explained above, the proposed set of benchmarks can be used to evaluate offline-RL algorithms as well as hybrid-RL algorithms, as it poses the problem of confounded datasets that do not have a standardized benchmark. For hybrid-RL algorithms, we use an imperfect simulator with transition error (as described in challenge 1), along with the dataset benchmarks described in this challenge.

## 4 EXPERIMENTS

In this section we present empirical evaluations following the procedues described in Section 3 above. We used online, offline, and hybrid RL methods to showcase challenges and limitations in current RL approaches for hybrid tasks. Our chosen array of methods represents a cross-section of current state-of-the-art RL approaches in both model-based and model-free paradigms, providing a broad look at how diverse techniques perform in the face of our hybrid RL benchmarks.

### 4.1 BASELINES

To test online-RL algorithms on the proposed simulators we used **TD3** (Fujimoto et al., 2018) and **SAC** (Haarnoja et al., 2018). To test offline-RL algorithms, we used the model-based (**MOPO** (Yu et al., 2020)), as well as state of the art model-free approaches (**TD3-BC** (Fujimoto and Gu, 2021) and **IQL**(Kostrikov et al., 2021)). Finally, to test hybrid-RL algorithms, we used two algorithms that can leverage both a simulator and offline data: the **H2O** (Niu et al., 2022) algorithm and a novel approach we term **HyCORL**. H2O adaptively pushes down or pulls up Q-values on simulated data according to the dynamics gap evaluated against real data. In what follows, we breifly describe HyCORL. We refer the reader to Appendix B for full details and implementation.

**HyCORL.** The **Hy**brid **C**orrection for **O**ffline **R**einforcement **L**earning is model-based, dynamics-aware, policy optimization algorithm. In this approach, we first train a correction function $f$ that learns to fix the discrepancy between observed data and the simulator's outputs. This is achieved by running each observation-action tuple $(o_i, a_i)$ through the simulator, and collecting its outputs (i.e., the simulator's computed next observations $o'_{i,\text{sim}}$). Then, the correction function's goal is to learn an additive function that fixes the gap between the simulator's next observation and the next observation registered in the dataset. This is done by minimizing the following loss: $\mathcal{L}(f) = \frac{1}{N} \sum_i \|o'_i - (o'_{i,\text{sim}} + f(o_i, a_i))\|$, where $N$ is the number of observations in the dataset. We note that in the worst case, when the simulator is completely incorrect, the correcting function should learn to output $o'_{i,\text{sim}} - o'$, which would typically be as difficult as learning the transition function directly from data (i.e., learning a model that outputs the next state given the current state and action) as seen in other offline-RL algorithms such as MOPO.

To train the agent, we first initialize the state by randomly selecting one from the given dataset. Then the transition function, which consists of a simulator and a correction function, is used to determine the next observations up to a predetermined horizon. Finally, the reward is penalized by the amount of uncertainty in the transition model evaluations. These last steps are similar to the MOPO algorithm, with the difference that HyCORL can combine the given dataset with a given simulator. See Appendix B for more details.

### 4.2 RESULTS

We demonstrate the performance of the different RL approaches on the challenges described in Section 3. We benchmarked the challenges detailed above on the MuJoCo-HalfCheetah environment, and present here the main highlights and insights that stem from these experiments. In the main text we present results for challenges 1, 2 and 4. Full details and more results, including results for challenge 3 and results on other environments, can be found in Appendix C. We report results as mean and standard deviation of the normalized rewards (normalization scales the raw rewards to a scale of 0 (random) to 100 (expert) as suggested in D4RL) across three random seeds.

In Figure 3 we show how different RL approaches perform on the modeling error challenge. Both Hybrid-RL algorithms, HyCORL and H2O demonstrate an interesting phenomenon. First, as expected,

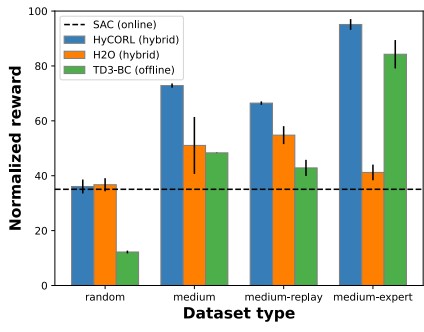 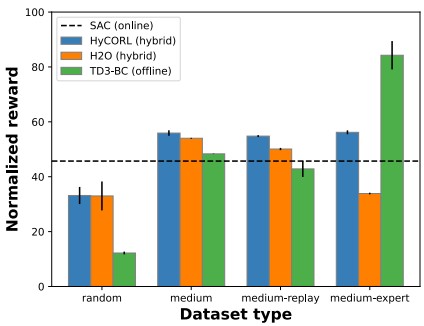

(a) Simulator with transition error          (b) Partially observed simulator

Figure 3: Results on HalfCheetah environment for modeling error and partial observability. In both figures, the algorithms have access to the standard D4RL datasets, but use different types of imperfect simulators. For modeling error **(a)** we introduced an error in the transition function by setting the gravitational parameter to $g = 19.6$ instead of $9.81$, and for partial observations **(b)** we added Gaussian noise ($\sigma = 0.05$) to the full state.

on the medium and medium-replay datasets both methods score better than SAC (online-RL), which uses only the simulator, and TD3-BC (offline-RL) which uses only the datasets. However, when using the simulator with observation error and the random dataset, we observed both hybrid-RL algorithms scored *worse* than only using SAC on the simulator – unexpectedly, using the offline dataset negatively impacted the hybrid approaches. We observed the same phenomenon in other cases as well. For example, in the medium-expert dataset with a partially observable simulator, HyCORL scored less than TD3-BC trained on the data alone, and H2O scored even worse, being inferior to both SAC on the simulator alone and TD3-BC on the dataset alone.

In Figure 4, we demonstrate the effect of hidden confounders in the dataset. We compare two different types of algorithms: an online-RL algorithm (SAC) on a partially observable simulator, and an offline-RL algorithm (TD3-BC) on the medium-expert dataset with hidden confounders. In the online case, the algorithm has access to the full state excluding a single dimension, and in the offline case, we remove the exact same variable from the dataset, despite the fact that it was used by the agent generating the dataset. Note that algorithms that do not use offline data cannot suffer from hidden confounding, though they might suffer from partial observability. We trained both algorithms across all possible variables, and compared the results. While one might expect the importance of a variable $v$ for performance in the online algorithm to be similar to its importance for the offline learning, we demonstrate that some variables are more important in the offline case. For example, `pos-root-z` (the $z$ coordinate of the front tip), has significant effect in offline TD3-BC, whereas `v-root-x` (the $x$ coordinate velocity of the front tip) significantly affected online SAC. This suggests that variable `pos-root-z` induces strong hidden confounding, significantly affecting the reward as well as the choice of actions by the data-generating-agent.

In Table 1 we show performance of all methods on HalfCheetah with hidden confounding error. For the offline datasets we used D4RL and applied confounding error. To achieve this we removed two variables from the dataset, even though the data-generating policy was dependent on them. For low confounding, $h_{\text{low}}$ corresponds to a variable of low confounding error ($\omega$-`front-foot`) and $h_{\text{high}}$ corresponds to high confounding error (`v-root-z`).

For the online simulator we used a simulator with transition error in the gravitational parameter ($g = 19.6$ instead of $9.81$). Under low confounding, HyCORL scored best across all options except the random dataset, where the simulator alone performed slightly better. Under high-confounding, both hybrid models and MOPO suffered severely. Interestingly, on the medium-expert dataset, which is twice as big as the medium dataset and has access to *optimal* trajectories, these algorithms' scores diminish, emphasizing the negative effects of hidden confounders in the data even on hybrid methods.

Overall, while the expectation is for hybrid reinforcement learning (RL) algorithms to perform at least as well as the best between online and offline approaches, our results reveal that in several cases

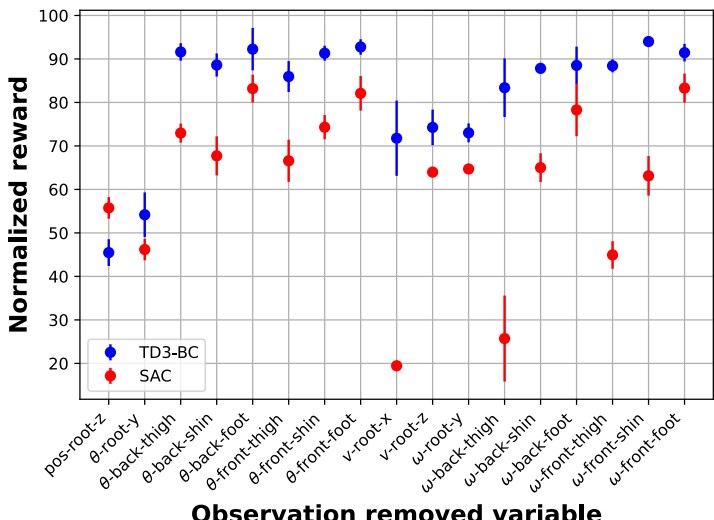

Figure 4: Results of offline (TD3-BC) and online (SAC) algorithms on the HalfCheetah environment with a single missing variable. TD3-BC runs on the medium-expert dataset. For each label on the x-axis, SAC trained on partially observed simulator that lacks that variable, and TD3-BC trained on a dataset that did not have any information about that variable, despite it being used by the agent which generated the dataset.

Table 1: Normalized reward on HalfCheetah environment, on four types of datasets, all with confounding errors. Online and Hybrid models also have access to a simulator with transition error in the gravitational parameter.

| Dataset | Conf. | Online-RL (sim) | | Offline-RL (data) | | | Hybrid-RL | |
|---|---|---|---|---|---|---|---|---|
| | | TD3 | SAC | MOPO | TD3-BC | IQL | H2O | HyCORL |
| Random | $h_{low}$ | $35.3 \pm 1.9$ | $35.1 \pm 2.4$ | $37.4 \pm 1.0$ | $11.7 \pm 0.6$ | $12.4 \pm 3.0$ | $34.2 \pm 1.1$ | $31.5 \pm 3.3$ |
| | $h_{high}$ | | | $26.2 \pm 3.3$ | $9.0 \pm 0.9$ | $6.6 \pm 3.0$ | $31.0 \pm 2.1$ | $30.1 \pm 0.4$ |
| Medium | $h_{low}$ | $35.3 \pm 1.9$ | $35.1 \pm 2.4$ | $60.6 \pm 7.1$ | $48.2 \pm 0.2$ | $48.4 \pm 0.2$ | $54.3 \pm 2.5$ | $75.2 \pm 2.6$ |
| | $h_{high}$ | | | $29.4 \pm 4.1$ | $46.1 \pm 0.5$ | $46.5 \pm 0.1$ | $34.5 \pm 3.4$ | $35.2 \pm 1.7$ |
| Medium replay | $h_{low}$ | $35.3 \pm 1.9$ | $35.1 \pm 2.4$ | $58.7 \pm 8.0$ | $44.6 \pm 0.3$ | $43.8 \pm 1.1$ | $49.9 \pm 4.9$ | $66.5 \pm 0.8$ |
| | $h_{high}$ | | | $32.9 \pm 1.1$ | $41.6 \pm 1.9$ | $42.5 \pm 0.0$ | $22.7 \pm 7.5$ | $37.1 \pm 1.4$ |
| Medium expert | $h_{low}$ | $35.3 \pm 1.9$ | $35.1 \pm 2.4$ | $52.7 \pm 4.4$ | $91.4 \pm 2.0$ | $90.7 \pm 3.0$ | $34.3 \pm 7.7$ | $99.3 \pm 0.8$ |
| | $h_{high}$ | | | $2.9 \pm 0.8$ | $74.3 \pm 4.1$ | $64.0 \pm 3.4$ | $18.7 \pm 4.7$ | $27.0 \pm 2.0$ |

this is far from reality. Moreover, we identify hidden confounding as a significant issue, markedly influencing performance of offline methods.

## 5 CONCLUSIONS AND FUTURE WORK

In this paper we provide some insights into the challenges encountered when combining offline data with imperfect simulators in RL. Our newly introduced B4MRL benchmarks facilitate the evaluation and understanding of such complexities, highlighting four main challenges: simulator modeling error, partial observability, state and action discrepancies, and confounding bias.

Our introduced hybrid algorithm, HyCORL, has proven beneficial. However, our results reveal that current hybrid methods combining simulators and offline datasets do not always lead to superior performance, pointing to an important future research direction. In addition, hidden confounders in the dataset can significantly affect the performance of all tested methods, including hybrid ones. In light of these results, we suggest future work to focus on developing more robust hybrid RL algorithms that can better handle modeling errors and hidden confounders, and that perform as least as well as either simulator based methods or offline learning. We believe the benchmarks and challenges proposed in the paper can help the community make strides towards more reliable and efficient RL models.

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
