# OpenReview forum: "Benchmarks for Reinforcement Learning with Biased Offline Data and Imperfect Simulators"
_ICLR.cc/2024/Conference — Submitted to ICLR 2024_

### Official Review · Reviewer_Q9WM · 2023-10-13

**Soundness:** 2 fair
**Presentation:** 2 fair
**Contribution:** 2 fair
**Rating:** 3
**Confidence:** 3

**Summary:**

The paper proposes a set of "Benchmarks" for Offline Reinforcement Learning. Assuming that the RL practitioner can be equipped with some data about the environment plus an imperfect simulator of the "world", the authors identify four main ways in which the data and simulator can "confound" the RL agent, resulting in suboptimal policies. Guided by those four "challenges" for hybrid offline RL approaches, the authors generate synthetic datasets and prepare custom configurations using the Mujoco simulator. They also provide some analysis of how those four aspects affect some standard learning RL methods.

**Strengths:**

- There is value in the identification of the main challenges that an RL agent might encounter, especially in the identification of the "Confounding bias", which is not as trivially thought of as the other challenges.
- If made freely-available, the datasets/implementations might assist in the development of new hybrid offline+online RL methods.
- The authors have a comprehensive view of all the resources an RL designer might get access to and consider it in their modeling.

**Weaknesses:**

- The main problem to me is that the work proposed is too incremental for a main track paper at ICLR. While there is value in the identification of the main challenges faced by the RL agent in the offline data+simulator scenario, it is a very incremental step from already-existing datasets/works, and procedurally, generating those benchmarks are relatively easy given Mujoco is a simple to execute and easily-available environment. For instance, someone interested in generating a similar benchmark might not even have to run the environment too many times by leveraging already-existing datasets. Starting from (Fu et al, 2020) datasets, one could modify the datasets by simply (i) changing an state variable; (ii) omitting a recorded state variable (iii) adding noise to the recorded action; (iv) adding noise to observation and action, to replicate a similar dataset generation as the one done by the authors, maybe 2 or 3 of the generated configurations only would require running all the experiment again (and even those, represent a small change in the Mujoco environment).

- A lot of work can be done to improve clarity of the work as well. It's not very clear how the datasets replicate the "structure" in Figure 1. The offline data and simulator construction is explained, but how is the agent performance validated? Is there a ground truth environment that is not modified from which performance measures are extracted?

- While the authors' descriptions make sense, there is not enough evidence that the four challenges identified by the authors actually represent what happens in a real environment. I strongly recommend the authors to include a more complex and less-controlled environment than Mujoco, doing the actual work of gathering data and building a simulator for the environment, and showing that those challenges actually represent practical reality. Alternatively, a domain where there are simulators of different fidelities could be used, that way the authors wouldn't need to carry out the work of developing a new simulation.

- I was surprised the authors didn't mention the very related multi-fidelity MDP formulation. The formulation models exactly one of the problems the authors are coping with, imprecise simulators.

Silva, F. L., Yang, J., Landajuela, M., Goncalves, A., Ladd, A., Faissol, D., & Petersen, B. (2023). Toward Multi-Fidelity Reinforcement Learning for Symbolic Optimization. Adaptive and Learning Agents (ALA) Workshop.

- In order to strengthen the contribution of the paper I suggest the authors follow one of the avenues:
1) Develop a new Hybrid algorithm where you can show clear performance improvement across most of the benchmarks
2) Introduce a more complex environment to the benchmarks.

**Questions:**

- Will all the implementations and datasets be made publicly available if the paper is published?
- how is the agent performance validated? Is there a ground truth environment that is not modified from which performance measures are extracted?

---

> ### Author Response · Authors · 2023-11-21
>
> Thank you for the feedback.
>
> **Regarding weaknesses:**
> 1. While some of our benchmarks might seem easy to implement, we believe that our contribution does not stop at the technical implementations of these benchmarks, but in the formulation of the benchmark as a whole and in identifying the distinct challenges associated with hybrid modeling. To this day, there is no standardized way to benchmark hybrid-RL algorithms, and indeed we believe we are the first to even outline the issues at play. We hope that our benchmark could provide researchers a starting point when crafting new hybrid algorithms, and inspire novel research in that area.
>
> 2. We are sorry for the confusion. In all experiments, the ground truth environment is the unbiased simulator. For example, in the modeling error experiment, the training agents have access to the biased simulator (e.g., with erroneous gravitational parameter) and some form of offline data. The offline data was collected from the ground truth environment but suffers from one of the biases of offline data. To test the algorithm, we evaluate it’s capabilities on the unbiased simulator (e.g., with the correct gravitational parameter). We will clarify this point in the text.
>
> 3. Regarding limitations of experiments. We agree that having more experiments could be beneficial to our benchmarks, and we will add more with time.
>
> 4. Thank you for the helpful feedback. We will add this relevant paper to our revised version.
>
> **Regarding questions:**
> 1. The entire code we used for dataset generation, benchmarks, and algorithms will be made available. Our goal is that researchers will be able to use our benchmarks as easily as possible.
> 2. See point 2 above.

---

### Official Review · Reviewer_hLBa · 2023-10-17

**Soundness:** 2 fair
**Presentation:** 3 good
**Contribution:** 1 poor
**Rating:** 3
**Confidence:** 3

**Summary:**

This paper considers challenges in combining offline data and imperfect simulators in reinforcement learning. The authors first summarize and explain the four major aspects: modeling error, partial observability, state and action discrepancies, and hidden confounding. The main contribution of the paper is a systematic way to create benchmarks for evaluating RL algorithms under the above 4 challenges. Finally, the authors evaluate popular existing methods on the benchmark environments, together with a new approach the authors proposed to combine imperfect simulator with biased data.

**Strengths:**

The writing is clear and presentation of this paper is well organized. It contains good summaries on the challenges which could provide the general audience with valuable background materials.

**Weaknesses:**

The exact contributions of this paper seem limited and insignificant. While it provides a good context in terms of the background, the precise benchmarks and experiments do not seem to offer new innovations/insights. Most of the environment modifications for the proposed challenges seem to be either adding noise or masking some dimensions, which probably have been used in previous work except not being formulated precisely as benchmarks. The experiments are also very limited on HalfCheetah and I'm not convinced why would this paper be a good benchmarking paper that the RL community can benefit from.

- This paper might benefit by enlarging the experimental space with diverse environments and a more comprehensive evaluation of algorithms to see if new insights can be drawn.
- For some environment modifications, the authors mention that "the choice of values was done based on the results of the SAC algorithm". Could this introduce in certain bias if solely based on one algorithm?

**Questions:**

See above.

---

> ### Author Response · Authors · 2023-11-21
>
> Thank you for the feedback.
>
> Regarding contribution:
> While some of our benchmarks might seem easy to implement, we believe that our contribution does not stop at the technical implementations of these benchmarks, but in the formulation of the benchmark as a whole and in identifying the distinct challenges associated with hybrid modeling. To this day, there is no standardized way to benchmark hybrid-RL algorithms, and indeed we believe we are the first to even outline the issues at play. We hope that our benchmark could provide researchers a starting point when crafting new hybrid algorithms, and inspire novel research in that area.
>
> Regarding limitations of experiments. We agree that having more experiments could be beneficial to our benchmarks, and would add more with time.
>
> Regarding SAC algorithm:
> The calibration of our algorithms was indeed based on the SAC algorithm, which as you point out can introduce some bias to the noise levels chosen. However, as seen in the experiments section, the chosen noise levels affect other baselines similarly to SAC.

---

### Official Review · Reviewer_RmHk · 2023-10-23

**Soundness:** 1 poor
**Presentation:** 1 poor
**Contribution:** 1 poor
**Rating:** 3
**Confidence:** 4

**Summary:**

This paper studies the performance of various (offline) RL methods when biased offline logged data and imperfect simulators are available. The paper also proposes a hybrid method, which uses both simulator and offline data to learn a new policy, and shows its benefits over existing offline RL methods (which only use offline data without simulation) in D4RL benchmarks. The paper publicizes the imperfect simulator used in the experiment as "Benchmarks for Mechanistic Offline Reinforcement Learning (B4MRL)".

**Strengths:**

1. **Considering the practical situation of having an (imperfect) simulator and offline logged data.** The setting this paper considers is reasonable and can be practically relevant in applications where simulators are often used for sim2real transfer (e.g., robotics).

2. **Publicizing a benchmark suite for future research.** Providing a benchmark is beneficial not only for reproducible research but also for facilitating future research on relevant topics.

3. **Highlighting challenges in using simulation and offline datasets.** Specifying challenges of using imperfect simulator and offline logged data can be informative for readers.

**Weaknesses:**

1. **The focus of this paper (either benchmarking or proposing a new method) is not clear.** It was unclear to me if this paper focuses on benchmarking or proposing a new hybrid method to leverage simulators and offline logged data. If this paper focuses on benchmarks, more details of the design choices should be discussed, e.g., why specific RL environments are chosen for the benchmark tasks or why specific parameters are chosen for the configurations of imperfect simulators and offline data. In contrast, if the paper focuses on proposing a new hybrid method, a more detailed discussion of existing work is needed, and logic behind why the proposed method works (e.g., theoretical analysis) should be explained. Of course, a research paper can have both benchmarking and method proposal as two contributions of the paper, however, this paper lacks the details in both contributions.

2. **Lacks the theoretical analysis of the four challenges.** The paper describes the intuition of the four challenges in using a simulator and offline data, however, its influence on the performance is unclear. For example, providing the upper bound of the performance decrease caused by the modeling error of the simulator and that caused by the bias of logged data can be informative for the readers. The manuscript is ambiguous about how the simulator and offline logged data compensate for errors with each other.

3. **Lacks the discussion of related work.** From the manuscript, I could not figure out if the proposed method is the only approach trying to combine the use of a simulator and offline data due to the lack of discussion of existing approaches. In addition, the advantage of the proposed method over model-based offline RL approaches was not discussed. For example, is the proposed method more effective than a representative model-based offline RL method proposed (e.g., COMBO (Yu et al., 21)) and why the proposed method performs better?

Yu et al, 21: Tianhe Yu, Aviral Kumar, Rafael Rafailov, Aravind Rajeswaran, Sergey Levine, Chelsea Finn. "COMBO: Conservative Offline Model-Based Policy Optimization". NeurIPS, 2021.

4. **The presentation of the paper should be improved.**

**Questions:**

1. What kind of errors do the simulator and offline data compensate for each other?

2. What are the advantages of the proposed method over existing model-based offline RL methods?

---

> ### Author Response · Authors · 2023-11-21
>
> Thank you for the valuable feedback.
>
> **Regarding Weaknesses:**
> 1. The main focus of this paper (“Benchmarks for Reinforcement Learning with Biased Offline Data and Imperfect Simulators”) is to introduce a novel benchmark for algorithms that can leverage biased simulators and offline data. However, currently there are not many algorithms that can handle this case, as there are mainly two types of algorithms: online ones that can handle sim2real gaps, and offline ones that handle biases in the data. In order to make our benchmark more complete, we decided to create a new algorithm which is a natural extension of the widely used MOPO (Yu et. al. Model based offline policy optimization) to the hybrid case. As we show in our experiments, most of the time the new algorithm can leverage the two forms of information (online and offline), but in some cases it fails. We believe that this benchmark, and the results that we demonstrate, could inspire researchers in crafting new and more advanced hybrid algorithms.
> 2. We agree that a theoretical analysis could make a valuable addition to understanding this problem. However, we believe that as a benchmark paper, developing novel theoretical bounds is beyond the scope of this specific work. We hope that our outlining of the challenges of the hybrid setting will spur future theoretical analyses.
> 3. As we mention in section 4.1, to the best of our knowledge there is currently only a single hybrid algorithm (except the one we proposed), which is named H2O. Regarding HyCORL vs. MOPO, as we write in section 4.1 and in appendix B (which can be found in the supplementary material), the only difference between HyCORL and MOPO is that HyCORL can utilize a given simulator. It does so by training the transition function to fix the simulator's outputs instead of learning the transition entirely from data. We will clarify this point in the main text.
>
>
>
> **Regarding questions:**
> 1. There are two potential sources of error compensation: The simulator is not affected by confounding errors and can be used extensively for exploration. The offline data was gathered in the real world so it does not suffer from errors in the model (transition errors, action errors etc.).
> Many current research efforts are made to either build an online algorithm that can overcome simulator errors, or to build an offline algorithm that is less affected by data biases. A hybrid algorithm should be able to balance between the disadvantages of both sources of information. For example, HyCORL was able to use D4RL offline data to fix an erroneous simulator (which has wrong parameters for example), and learn a very reliable transition function, which was then used to achieve better rewards.
> 2. As mentioned above, the only advantage HyCORL has over MOPO is that it can utilize an online simulator, even if it is biased. In case the simulator simulates the environment perfectly, HyCORL could learn to use the simulator's predictions and not use data at all. When the simulator is highly noisy or biased, HyCORL should collapse to produce results such as MOPO (as mentioned in section 4.1 – HyCORL).

---

> > ### Comment · Reviewer_RmHk · 2023-11-22
> >
> > Thank you for the response. After carefully reading it, I will maintain my initial score. I still have concerns about the impact of this paper to the community due to the weaknesses I pointed out in the initial review.

---

### Official Review · Reviewer_nns3 · 2023-10-28

**Soundness:** 3 good
**Presentation:** 2 fair
**Contribution:** 3 good
**Rating:** 6
**Confidence:** 3

**Summary:**

This work studies the hybrid RL setting that combines offline datasets with simulators. It first outlines four challenges pertaining to offline2real and sim2real, together with some brief descriptions of real-world examples: (i) simulator error, (ii) state discrepancy, (iii) action discrepancy, (iv) unobserved confounding.

It then presents a set of benchmarks modified from D4RL-mujoco to exhibit each of the four challenges. Benchmarking results are provided for purely simulator-based online RL, purely dataset-based offline RL, and hybrid approaches that make use of simulator and dataset.

**Strengths:**

- The hybrid RL problem studied by this paper has high practical importance.
- The proposed procedures to induce the practical challenges are systematically presented, and could be generalizable to different environments.

**Weaknesses:**

- Result section could benefit from some organization to bold/highlight the key takeaways of different sub-experiments.For example, what are some general trends in terms of when hybrid methods perform better than simulator-based or offline RL?

**Questions:**

- The benchmark description says both mujoco and Highway environments are used. Where is the results for Highway?

---

### Meta-Review · Area_Chair_JQhR · 2023-12-11

**Metareview:**

This paper proposes addressing four challenges in combining offline data with imperfect simulators in RL: modeling error, partial observability, state and action discrepancies, and hidden confounding. To drive progress in the field, the authors introduce Benchmarks for Mechanistic Offline Reinforcement Learning (B4MRL) and present a new approach that combines an imperfect simulator with biased data, demonstrating its efficacy. There are some weaknesses of the paper raised from the review comments and discussions, including the incremental technical novelty, weak support of theoretic analysis, insufficient related work discussions, and the presentation. Although the authors provided detailed feedbacks, some of the concerns raised are still unsolved.

**Justification For Why Not Higher Score:**

There are some weaknesses of the paper raised from the review comments and discussions, including the incremental technical novelty, weak support of theoretic analysis, insufficient related work discussions, and the presentation. Although the authors provided detailed feedbacks, some of the concerns raised are still unsolved.

**Justification For Why Not Lower Score:**

N/A

---

### Decision · Program_Chairs · 2024-01-16

Reject